# Integrated IBD Analysis, GWAS Analysis and Transcriptome Analysis to Identify the Candidate Genes for White Spot Disease in Maize

**DOI:** 10.3390/ijms241210005

**Published:** 2023-06-11

**Authors:** Dong Wang, Yue He, Lei Nie, Shuang Guo, Liang Tu, Xiangyang Guo, Angui Wang, Pengfei Liu, Yunfang Zhu, Xun Wu, Zehui Chen

**Affiliations:** 1College of Agriculture, Guizhou University, Guiyang 550006, China; wdong926400@163.com (D.W.); 13985628529@163.com (Y.H.); nielei1812@163.com (L.N.); m18985987996@163.com (S.G.); 2Institute of Upland Food Crops, Guizhou Academy of Agricultural Sciences, Guiyang 550006, China; tudongyang1011@163.com (L.T.); xyguo0372@163.com (X.G.); wangangui123@163.com (A.W.); ggfp123@163.com (P.L.); wangtianyu139@sina.com (Y.Z.); 3Ministry of Agriculture and Rural Affairs Key Laboratory of Crop Genetic Resources and Germplasm Innovation in Karst Region, Guiyang 550006, China

**Keywords:** maize (*Zea mays* L.), identity-by-descent segments, genome-wide association study, transcriptome analysis, white spot disease

## Abstract

Foundation parents (FPs) play an irreplaceable role in maize breeding practices. Maize white spot (MWS) is an important disease in Southwest China that always seriously reduces production. However, knowledge about the genetic mechanism of MWS resistance is limited. In this paper, a panel of 143 elite lines were collected and genotyped by using the MaizeSNP50 chip with approximately 60,000 single nucleotide polymorphisms (SNPs) and evaluated for resistance to MWS among 3 environments, and a genome-wide association study (GWAS) and transcriptome analysis were integrated to reveal the function of the identity-by-descent (IBD) segments for MWS. The results showed that (1) 225 IBD segments were identified only in the FP QB512, 192 were found only in the FP QR273 and 197 were found only in the FP HCL645. (2) The GWAS results showed that 15 common quantitative trait nucleotides (QTNs) were associated with MWS. Interestingly, SYN10137 and PZA00131.14 were in the IBD segments of QB512, and the SYN10137-PZA00131.14 region existed in more than 58% of QR273′s descendants. (3) By integrating the GWAS and transcriptome analysis, Zm00001d031875 was found to located in the region of SYN10137-PZA00131.14. These results provide some new insights for the detection of MWS’s genetic variation mechanisms.

## 1. Introduction

Maize (*Zea mays* L.) is cross-pollinated with strong heterosis [1], and the outstanding hybrids are inseparable from the foundation parents (FPs). Commonly, FPs have many excellent features, e.g., good plant architecture, high grain yield, wide adaptability, strong resistance and a high general combining ability. In addition, they are also widely used in breeding and are derived from other FPs [2,3]. For instance, many broadly used varieties or elite heterotic groups, such as Stiff Stalk (SS), Non-Stiff Stalk (NSS) and Iodent, were developed from a limited number of FPs and contributed prodigiously to modern increases in maize grain production [4]. Some lines derived from Iodent were widely utilized by Pioneer Corporation, including the representative elite line PH207 [5]. In addition, most inbred lines derived from PH207, i.e., PHH93, PHG29, PHG44, PHN82, PH24E and PHK42, are also commonly used in current maize breeding [6,7]. To clarify the formation of maize FP lines, Wu et al. [8] identified 111 identity-by-descent (IBD) segments of PH207, which were associated with days to tasseling and pollen shedding. These IBD segments are very important for maintaining the characteristics of FPs during maize improvement and domestication practices [9]. Generally, these IBD segments not only maintain the specific traits of the FPs, but also contain favorable genes that play a key role in trait improvement and maintenance of heterosis performance [10]. Dissecting the transmission mode and functions of the IBD segments could help us better understand the formation of the FPs and provide theoretical support to guide the breeding of commercial inbred lines through molecular-assisted selection.

At present, changes in maize diseases are occurring due to global warming, planting structure and cultivation pattern changes [11]. For example, new diseases appear suddenly, and some recurrent diseases continue to break out in special years and under specific circumstances [12]. In recent years, because of the outbreak of maize white spot (MWS), many susceptible varieties have exhibited reduced production or even failed, which has created new challenges and difficulties in corn production, especially in Southwest China. MWS, also known as *Phaeosphaeria* leaf spot (PLS), was first reported in India in 1965 [13] and was prevalent in Brazil and the United States from the 1980s to 1990s [14,15], severely damaging maize production in tropical and subtropical regions and causing yield losses reaching up to 60% in Brazil [16,17]. In 2021, large areas of corn production were affected by MWS in Yunnan, with yield losses in the range of 10–50% [18]. Unfortunately, there are still no special treatments to effectively control MWS. Breeding practices show that the most economical and effective method to control MWS is still the breeding and application of resistant maize hybrids. To evaluate and clarify the resistance level of different FPs to MWS, a panel containing the FP lines QB512, QR273 and HCL645 and relevant derivative lines were constructed to identify relevant characteristic IBD segments of each FP by using high-throughput single nucleotide polymorphism (SNP) markers. Then, we clarified the relevant functions of each IBD segment for MWS by using genome-wide association study (GWAS) methods combined with transcriptome analysis to identify the candidate genes for MWS. These results provide new insights into the genetic variation mechanism of MWS. In addition, this study also provides effective information for the future innovation of maize germplasm resistant to MWS.

## 2. Results

### 2.1. Analysis of Resistance to MWS

According to the MWS identification across three locations in Yunnan (Appendix A, Figure 1), a significant effect of genotypes × environment was observed for MWS severity (Table 1), indicating significant differences in MWS severity among the population, and high genetic variability within the population. The average disease index (DI) of B73 with high susceptibility (HS) to MWS was 87.66 among the 3 environments (Figure 1B), which was consistent with the previous study [19]. The elite line QB512 with high resistance (HR) to MWS had an average DI of 6.17 across 3 environments. A total of 7 of the 11 QB512-derived lines showed HR to MWS, and 4 descendants showed resistance (R) to MWS (Figure 1A). The average DI of QR273 was 14.20 among the 3 environments, showing R to MWS (Figure 1B), wherein 4 of the 34 derivative lines of QR273 showed HR to MWS, 13 descendants showed R to MWS, 2 descendants showed moderate resistance (MR) to MWS, 1 descendant showed susceptibility (S) to MWS, and 14 descendants showed HS to MWS (Figure 1A). In addition, most descendants of HCL645 showed HS to MWS (Figure 1A). A comparison showed that QR273 and QB512 had good resistance to MWS, while HCL645 was highly susceptible to MWS.

### 2.2. Excavation of the IBD Segments

The IBD segment analysis results are shown in Appendix A and Figure 2 and Appendix A. In total, 299, 318 and 311 IBD segments were individually identified among lines derived from the FPs QB512, QR273 and HCL645. Pairwise comparison showed that 52, 40 and 92 IBD segments were shared between QB512 and QR273, QB512 and HCL645, and QR273 and HCL645, respectively. Triple comparison showed that 18 IBD segments were common among these 3 groups. These IBD segments may play a key role in the formation of FP lines. Additionally, the longest IBD segment of QB512, with an interval of 119.54 Mb (22.65–142.19 Mb) and containing 174 SNPs, was detected on chromosome 4. QR273 had the longest interval of 36.11 Mb (29.53–65.64 Mb), containing 30 SNPs on chromosome 7. The interval size of the HCL645 IBD segment was 79.02 Mb (142.87–221.89 Mb), containing 111 SNPs on chromosome 2.

### 2.3. GWAS of MWS

The GWAS results showed that 73, 25 and 31 quantitative trait nucleotides (QTNs) (*p* < 0.001) were significantly associated with MWS under the environments of Shilin (Appendix A, Figure 3A and Appendix A), Mojiang (Appendix A, Figure 3B and Appendix A), and Wenshan (Appendix A, Figure 3C and Appendix A), respectively. Under the environment of Shilin, 73 QTNs related with MWS were detected, located on chromosomes 1, 2, 3, 4, 6, 7 and 8. Under the environment of Mojiang, 25 QTNs related with MWS were detected, located on chromosomes 1, 2, 3, 5, 6, 8 and 10. Under the environment of Wenshan, 31 QTNs related with MWS were detected, located on chromosomes 1, 2, 3, 4, 8, 9 and 10. Based on the average value of MWS DI among 3 environments, 30 QTNs related with MWS were identified, located on chromosomes 1, 2, 3, 5 and 8 (Appendix A, Figure 3D and Appendix A). Overall, a total of 15 common QTNs were found after the comparison of results under different environments (Table 2), wherein 3 common QTNs were co-identified under both environments of Shilin and Wenshan, with SNP of PZE-101040783, PZE-102031753 and SYN30108, respectively. Six common QTNs were co-identified under both environments of Shilin and joint environments, with SNP of PHM13619.5, SYN11249, SYN10891, SYN37674, ZM012464-0529 and SYN4935, respectively. Two common QTNs were co-identified under both environments of Mojiang and joint environments, with SNP of SYNGENTA14387 and PZE-103084298, respectively. One common QTN was co-identified under both environments of Wenshan and joint environments, with SNP of PZE-101151153. Three common QTNs were co-identified among the environments of Shilin, Wenshan and joint environments, with SNP of SYN10137, PZA00131.14 and PZE-108057528, respectively. Interestingly, QTNs of SYN10137 and PZA00131.14 were in the identical IBD region of QB512, with an interval size of 19.36 Mb (200.56–219.92 Mb) on chromosome 1 (Figure 4B). Meanwhile, the SYN10137 (200.56 Mb)-PZA00131.14 (203.39 Mb) region was detected in more than 58% of the QR273′s descendants (Figure 4A).

### 2.4. The Comparision of Quantitative Real-Time PCR (qRT-PCR) and Transcriptome Analysis

The identification results of the differentially expressed genes (DEGs) showed that 2104 genes were upregulated and 3625 genes were downregulated in B73 when compared with QR273 (Appendix A, Figure 5A). These DEGs were involved in five primary processes, including cellular processes, environmental information processing, genetic information processing, metabolism and organismal systems. Among these upregulated genes, 12 genes were related to peroxisome, 39 genes were related to plant hormone signal transduction, 24 genes were related to protein processing in endoplasmic reticulum, 16 genes were related to carbon fixation in photosynthetic organisms, and 33 genes were related to plant–pathogen interaction (Figure 5B). Among these downregulated genes, 19 genes were related to peroxisome; 74 genes were related to the mitogen-activated protein kinase (MAPK) signaling pathway-plant; 24 genes were related to RNA transport; 18 genes were involved in valine, leucine and isoleucine degradation; and 142 genes were involved in plant–pathogen interaction (Figure 5C). In addition, five genes were randomly selected for qRT-PCR. The expression trends of the five genes were consistent with the transcriptome analysis results, which indicated that the transcriptome data were reliable.

### 2.5. Combining the IBD Analysis, GWAS and Transcriptome Analysis to Identify Candidate Genes

In total, 11 candidate genes were tightly linked with these common QTNs detected among 3 environments. To further determine the candidate genes for MWS, the results of the GWAS, IBD analysis, analysis of resistance and transcriptome analysis were incorporated. Among the candidate genes co-identified under joint environments and other environments (Table 2), the relative expression levels of Zm00001d027619 (Figure 6C), Zm00001d031641 (Figure 6D) and Zm00001d032024 (Figure 6F) were significantly higher in B73 (HS) than in QR273 (R), and Zm00001d041844 was upregulated in QR273 (Figure 6E). In addition, these four genes are involved in different physiological and biochemical processes (Figure 6A). Zm00001d027619 encodes beta-amylase. Zm00001d031641 encodes a membrane fusion protein that is involved in endocytosis and exocytosis. Zm00001d032024 encodes an myb transcription factor. Zm00001d041844 encodes methyltransferase. Among the candidate genes co-identified under Shilin and Wenshan (Table 2), Zm00001d002549 was downregulated in QR273, while Zm00001d053432, involved in the photosynthesis process, was upregulated in QR273 (Appendix A).

Among the 15 common QTNs, SYN10137 and PZA00131.14 were co-identified under Shilin, Wenshan and joint environments. Zm00001d031860, Zm00001d031875 and Zm00001d031899 were identified within the region of SYN10137-PZA00131.14, in which Zm00001d031875 encodes H(+)/Pi cotransporter and its relative expression levels were significantly higher in B73 (HS) than in QR273 (R) (Figure 7A,C). Moreover, a significant difference in the MWS DI was found in the lines with and without the SYN10137-PZA00131.14 region (Figure 7B).

## 3. Discussion

### 3.1. Inheritance of MWS Resistance

QR273 is a tropical maize germplasm with good resistance, but its descendants exhibit varying degrees of resistance to MWS, with some showing HR and others HS. Derivatives with resistance are mostly improved descendants between QR273 and tropical germplasm, while those with susceptibility are mostly the result of crossing with temperate germplasm. The differential MWS response observed among QR273’s descendants may be attributed to multiple factors. Genetic variation between the different lines could contribute significantly to differences in MWS resistance, as some ancestors may carry genes or gene combinations that reduced resistance to MWS, for example, HCL645, which is a temperate material with poor resistance. In the process of improving QR273 using temperate maize materials, due to different selection pressures, resistance was probably not emphasized in the selection of more resistant lines during breeding, or the resulting germplasm may have relatively poorer resistance due to a stronger genetic background from temperate maize material in the crossbreeding process [20]. Therefore, the descendants of QR273 showed a different performance to MWS.

### 3.2. Transmission of the IBD Segments

There is no doubt that FPs play a crucial role in maize breeding practices, and a few important FPs were used to construct many of the commercial inbred lines. For example, more than 70 inbred lines and 80 hybrids were derived from Huangzaosi (HZS) [21]. The QR273 derived from the Suwan 1 population was used to release more than 20 hybrids that have been widely used in maize production in Southwest China [22,23]. Meanwhile, there are more than 10 validated varieties with QB512 and a relevant improved line as their parents [24]. The dissection of the IBD segments and clarification of their transmission pattern would improve the understanding of FP formation. Wu et al. [25] performed a genome-wide scan of HZS and its descendants using the Maize SNP50 bead chip and identified 15 conserved genetic fragments, which were retained in more than 60% of the derivative lines of HZS. Liu et al. [26] detected 1262, 1373 and 1019 IBD regions in the derivative lines of Dan 340, Mo17 and HZS, respectively. In general, these IBD fragments contain many important QTNs. Using the derivative lines developed from the FPs, i.e., B73, Mo17, PH207 and HZS, Wu et al. [8] identified many IBD segments related to multiple agronomic traits, e.g., plant height (PH), ear height (EH), 100-kernel weight (KWE), ear row number (ER), kernel number per row (KNPR), ear length (EL), days to tasseling, anthesis and silking interval. Although some preliminary studies were performed on the genotypic characteristics of the FPs and their derivative lines and explored some regions controlling important traits, there are still many characteristic segments and their related functions that remain unidentified and need to be further explored and verified.

In this study, a total of 299, 318 and 311 IBD segments were detected in the descendants of QB512, QR273 and HCL645, respectively. With public databases, a total of 62 genes were identified within the longest IBD region of QB512, including several favorable genes (Appendix A). For instance, Zm00001d049905 belongs to the ABC transporter family and is associated with ATPase activity (https://www.uniprot.org/ (accessed on 1 May 2023)), which was detected in the longest IBD segment of QB512 and was tightly linked to many QTNs associated with EL [27], ear diameter [28], ER [29], KWE [30], seed desiccation [31], ear rot disease [32], southern rust [33] and MWS [34]. In addition, seven genes were located in the longest IBD segment of QR273 (Appendix A), wherein Zm00001d019677 encodes an f-box protein (https://www.uniprot.org/ (accessed on 1 May 2023)). Previous studies found that the longest IBD segment of QR273 was tightly linked to many QTNs for seed desiccation [31], ear rot disease [35] and MWS [36]. Forty-nine genes were in the longest IBD segment of HCL645, and they have functions in important physiological metabolic processes (Appendix A). For instance, Zm00001d005347 encodes monodehydroascorbate reductase homolog 1 (https://www.uniprot.org/ (accessed on 1 May 2023)). Many related reports have stated that this region is associated with several QTNs of essential traits, e.g., PH, EH, anthesis stage and seed desiccation [37,38]. The pedigrees of QB512 and QR273 include tropical materials, which have good resistance to MWS, but frustratingly, have long growth periods and poor seed desiccation. Differently, HCL645 belongs to the Iodent germplasm, which has shorter growth periods and fast seed dehydration, but is susceptible to MWS. Therefore, these IBD segments can be used as key regions for the improvement of temperate and tropical germplasm. By genome-wide selection, these IBD segments can be effectively fused to aggregate the beneficial traits of temperate and tropical germplasms and create excellent lines with temperate and tropical pedigrees in the future.

### 3.3. Specific Pathways Response to Diseases in Maize

The KEGG showed that the DEGs were significantly enriched in ‘plant-pathogen interaction’, ‘plant hormone signal transduction’ and ‘starch and sucrose metabolism’, which act in adversity for plants. For example, ‘starch and sucrose metabolism pathway’ is important for providing energy and carbon skeletons for various processes in plants. Moreover, the pathway correlated with improved stress tolerance [39,40], in which the expression of genes related to starch and sucrose metabolism in maize were induced by aluminum stress, leading to an increase in sucrose content and starch degradation, thereby enhancing maize tolerance to aluminum stress [41].

In the study, Zm00001d027619 is involved in starch and sucrose metabolism and encodes beta-amylase. Additionally, the relative expression levels of Zm00001d027619 were significantly higher in B73 (HS) than in QR273 (R). It has been reported that beta-amylase genes act as negative regulators of disease resistance in plants [42]. The pathogens maybe disrupt the enzyme systems involved in starch and sucrose metabolism, resulting in accelerated amylosynthesis and reduced sucrose used to resist MWS.

### 3.4. Integrating GWAS and Transcriptome Analysis to Reveal the Function of IBD Segments in MWS Resistance

According to the regularity of maize disease occurrence, it is predicted that MWS will gradually become an important disease in the southwest and even other regions [43]. Currently, domestic resistant germplasms with resistance to MWS are scarce, and more new elite lines and hybrids need to be quickly created to ensure the safe production of maize. Previously, some quantitative trait loci (QTLs) tightly linked to MWS were identified in bin 1.05/06, bin 2.06/07, bin 4.07/08, bin 6.02, bin 7.01, bin 8.05 and bin 8.07/8.08 using different populations and methods [34,36,44,45,46]. However, the number of QTLs was relatively small, and no high-density genetic regions have yet been identified that can be used as targets for gene map cloning and molecular-assisted breeding. Guo et al. [47] identified some IBD segments of QR273 and HCL645 using FPs, i.e., HCL645, Ki32, QR273 and PH4CV, but the SNPs and materials were not identical to those used in this paper. Additionally, the IBD region related to the MWS was not revealed.

In this paper, a total of 15 common QTNs were found to be significantly associated with the maize reaction to white spot severity, and 11 candidate genes were screened (Table 2). Among these candidate genes, some important genes for plant stress resistance were found. For instance, Zm00001d032024 encodes the myb transcription factor-38, which is involved in plant resistance to environmental stress, e.g., drought, salt and temperature [48]. Among the 30 QTNs identified in the joint environments (Appendix A), 8 QTNs were located in the same bin or the vicinity of the QTLs mapped by other authors, e.g., PZE-108133242 corresponded to the genetic regions reported by Carson et al. [44] (bin 8.07/8.08) and Moreira et al. [45] (*ph8b*: bnlg1607-bnlg1823). A total of 7 QTNs, i.e., SYN29300, SYN29299, SYN29298, PZE-101151153, SYN10891, SYN10137 and PZA00131.14 on chromosome 1, were located in *qMWS1.06* (umc1590-bnlg615) mapped by Lana et al. [34], and more than 30 resistance gene analogs were located in this region. In this study, we also identified important candidate genes in this genetic region. For example, Zm00001d031660, which encodes domains of unknown function protein, may contribute to defense responses to pests and diseases in plants [49]. Zm00001d031641, which encodes a membrane fusion protein and downregulates in QR273, plays an important role in the plant defense response [50]. Meanwhile, there was a phenomenon of pleiotropy of resistance loci, with the resistance loci related to other diseases, such as northern corn leaf blight [51], southern corn leaf blight [52], gray leaf spot [53] and common rust [54], in the genetic region. Therefore, these genes can be used as important targets for disease-resistance gene mapping and cloning and molecular-assisted breeding to create multiresistant elite inbred lines and hybrids.

Among the 15 common QTNs, PZE-108057528, SYN10137 and PZA00131.14 were co-identified under Shilin, Wenshan and joint environments, wherein SYN10137 and PZA00131.14 were located in identical IBD regions of QB512, with an interval size of 19.36 Mb (200.56–219.92 Mb). Additionally, SYN10137-PZA00131.14 (200.56–203.39 Mb), which existed in more than 58% of QR273′s descendants, was detected. Three candidate genes were identified within the region of SYN10137-PZA00131.14, i.e., Zm00001d031860, Zm00001d031875 and Zm00001d031899. Zm00001d031860 encodes 7-hydroxymethyl chlorophyll, a reductase chloroplastic, which is related to leaf senescence [55]. Zm00001d031899 encodes malate dehydrogenase (NADP(+)), which is involved in the defense responses of plants under adversity [56]. Zm00001d031875 encodes the H(+)/Pi cotransporter, which is involved in inorganic phosphate uptake in green parts of plants and contributes to the supply of nutrients to the leaves of plants [57]. Interestingly, according to the DEG analysis, Zm00001d031875 was downregulated in QR273. Based on other research, this region was linked to the QTNs for KNPR [8], cob diameter [58] and PH [59]. The genetic fragment is clearly rich in favorable genes. Most of the inbred lines that did not inherit this IBD fragment (SYN10137-PZA00131.14), such as the descents of HCL645, had a higher MWS DI. Conversely, most inbred lines that did have this segment had a lower MWS DI. Therefore, SYN10137-PZA00131.14 can be used as a key genetic region for improving resistance to MWS.

## 4. Materials and Methods

### 4.1. Plant Materials

Exactly 1 panel of 143 inbred lines was used as material, including 48 lines derived from the FP lines QB512, QR273 and HCL645. These lines were divided into three FP groups, according to the pedigree information. Detailed information was provided in Appendix A.

### 4.2. Phenotypic Evaluation and Analysis

In June 2022, 143 inbred lines were planted in the cities of Wenshan (1368 m, 23.35° N, 104.25° E), Shilin (1833 m, 24.66° N, 103.38° E) and Mojiang (1292 m, 23.46° N, 101.67° E) in Yunnan Province, China, which showed the most serious incidence of MWS. The field design was characterized by a 4 m row length, 0.65 m row spacing, 0.2 m plant spacing and 2 replications. Additionally, to facilitate the spread of pathogens, the inbred lines highly sensitive to MWS were planted around the field. The upper and lower 3 leaves of the ear were evaluated 15–30 days after flowering, and then resistance grades were calculated to classify resistance levels (Table 3). Furthermore, according to the combined calculation results, the degree of MWS for each line was determined, and the DI was calculated for each line as follows: DI = ∑ (number of diseased leaves at each level × severity level at that level)/(total number of leaves surveyed × 9) × 100) (Table 4). Finally, descriptive statistics and analysis of variance (ANOVA) for the phenotype data were calculated and performed by using SAS 9.2 software.

### 4.3. Genotyping

The panel of 143 inbred lines was genotyped by using the MaizeSNP50 chip, including 60,000 SNPs. At the five-leaf stage [60], leaves of each line were sampled in bulk to extract genomic DNA by the improved CTAB procedure. The genotyping work was completed by the Beijing Compass Biotechnology Company according to the Infinium1 HD assay ultra protocol guide. All genotypes were integrated according to an identical physical position with the B73 reference genome (RefGen_v4). The obtained genotype data were filtered according to the following SNP filtering standards: (1) missing rate < 20%, (2) minor allele frequency (MAF) > 0.05 and (3) indistinct marker position on the physical map of the B73 reference genome (RefGen_v4). Finally, for the 143 inbred lines, 40,648 high-quality SNPs were obtained and used for the next analysis.

### 4.4. Identification of the IBD Segments

According to the pedigree of each inbred line, FPs and their derivatives were identified. To excavate IBD segments, derivatives from the same FP were compared with their FP synchronously, using the SNPs without heterozygous genotypes and missing genotypes. A total of 3157 SNPs were selected for genome-wide comparison to identify the IBD segments between the FPs (QB512, QR273 and HCL645) and relevant derived lines. If a genomic region was inherited concurrently by all the descendants of an FP and contained at least two SNPs or more, it was identified as an IBD [9].

### 4.5. Analysis of the Functions of the IBD Segments in MWS

In total, 40,648 SNPs were used in the GWAS, which was operated in the R 4.2.1 package GAPIT (https://www.r-project.org/ (accessed on 1 May 2023)) using the mixed linear model (MLM), with population structure and pair kinship as the covariates index to correct for false positive associations between individual loci and phenotypes. QTNs were selected when the *p* values of the SNPs were less than 0.001 [61]. Then, the overlap analysis was performed between the results of IBD segment detection and the GWAS, and stepwise regression and Student’s *t*-test were used to analyze the correlation between IBD segments and MWS resistance results. For each important genetic region, if one candidate gene predicted by the filtered gene set model was tightly linked with the tag SNPs, it was treated as tightly linked with target traits [62] and was selected for in-depth analysis.

### 4.6. Candidate Gene Prediction and Transcriptome Data Analysis

According to relevant public data (http://www.maizegdb.org/ (accessed on 1 May 2023)), biological information analysis was performed to explore the functions of candidate genes associated with MWS resistance. Based on the MWS resistance results, B73 and QR273 leaves were collected for total RNA extraction at the nine-leaf stage. RNA was collected by using TRIzol reagent, and the construction of cDNA libraries and RNA sequencing were performed by Biomarker Technologies (Beijing, China) with the Illumina HiSeq 2000 platform. By combining the relative expression and functional annotation data of the genes, candidate genes were identified [63]. Fragments per kilobase of transcript per million mapped reads (FPKM) values were used to evaluate gene expression levels. The DEGs were detected by using the R 4.2.1 statistical package DESeq with a standard procedure, including an adjusted *p* value (*P_adj_*) < 0.05 and a fold change ratio (|log_2_ [FC]|) ≥ 1 [64].

### 4.7. qRT-PCR Analysis

To verify the reliability of the transcriptome data, five genes were randomly selected for qRT-PCR. According to the method described by Wu et al. [65], the qRT-PCR was designed and performed. The qRT-PCR primers were designed by Primer 5.0 software and listed in Appendix A.

## 5. Conclusions

This study showed that SYN10137-PZA00131.14 was not only a multiple disease resistance locus, but also a genetic region associated with other agronomic traits. Thus, it is an important genetic region for the improvement of temperate and tropical maize germplasm, especially in the creation of new maize germplasm resistant to MWS. By integrating the IBD, GWAS and transcriptome analysis results, Zm00001d031875 was identified as an important candidate gene for MWS resistance.

## Figures and Tables

**Figure 1 ijms-24-10005-f001:**
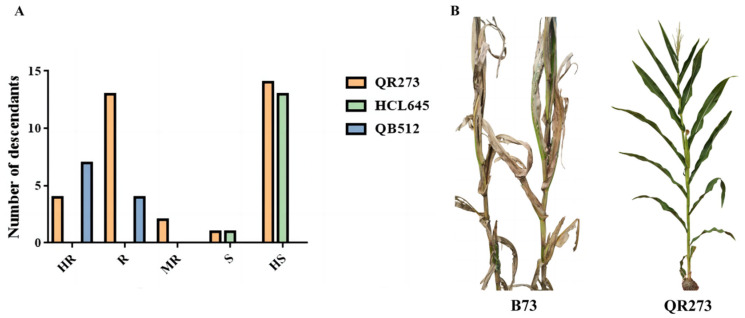
Identification results of MWS in Yunnan. Notes: (**A**) MWS response of derivative lines from QR273, HCL645 and QB512; (**B**) Response of B73 and QR273 to MWS.

**Figure 2 ijms-24-10005-f002:**
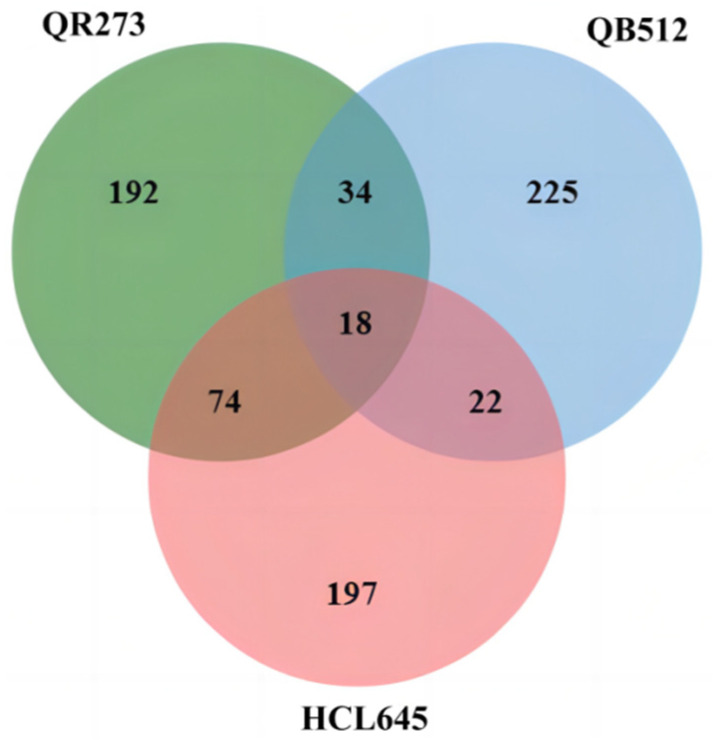
Venn diagram of IBD segments between QB512, QR273 and HCL645.

**Figure 3 ijms-24-10005-f003:**
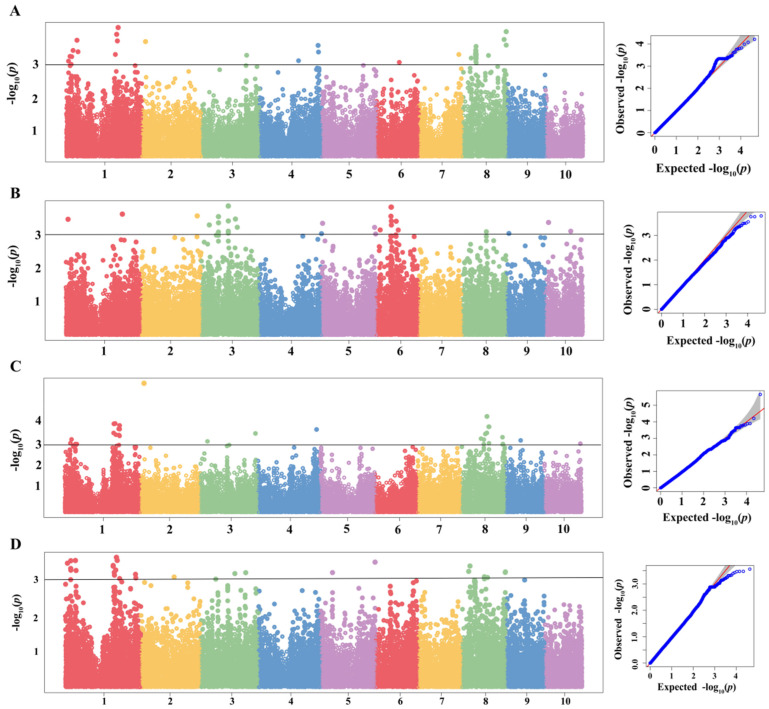
Plots of GWAS results for MWS under different environments by using population structure and pair kinship as covariates index to correct for false positive associations between individual loci and phenotypes. The significance threshold was −log_10_ (*p*) > 3.0 for the Manhattan plot (left). The quantile–quantile (Q-Q) plot (right), which demonstrated the overlapped and exceeded associations between the observed signals (blue dots) and the expected (red lines) signals under the null hypotheses. Notes: (**A**) The Manhattan plot and QQ plot, displaying 73 QTNs associated with MWS (*p* < 0.001) in Shilin; (**B**) The Manhattan plot and QQ plot, displaying 25 QTNs associated with MWS (*p* < 0.001) in Mojiang; (**C**) The Manhattan plot and QQ plot, displaying 31 QTNs associated with MWS (*p* < 0.001) in Wenshan; (**D**) The Manhattan plot and QQ plot, displaying 30 QTNs associated with MWS (*p* < 0.001) in joint environments.

**Figure 4 ijms-24-10005-f004:**
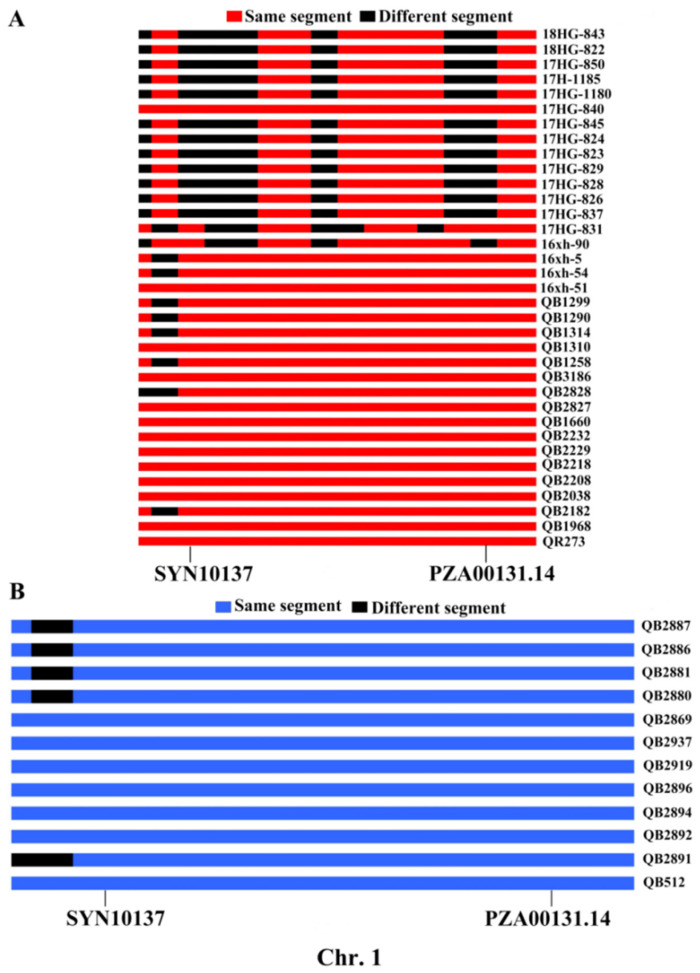
Transmission of the region (SYN10137-PZA00131.14) located on Chr 1 among the descendants of QB512 and QR273. Notes: (**A**) Transmission of the region (SYN10137-PZA00131.14) among the descendants of QR273; (**B**) Transmission of the region (SYN10137-PZA00131.14) among the descendants of QB512.

**Figure 5 ijms-24-10005-f005:**
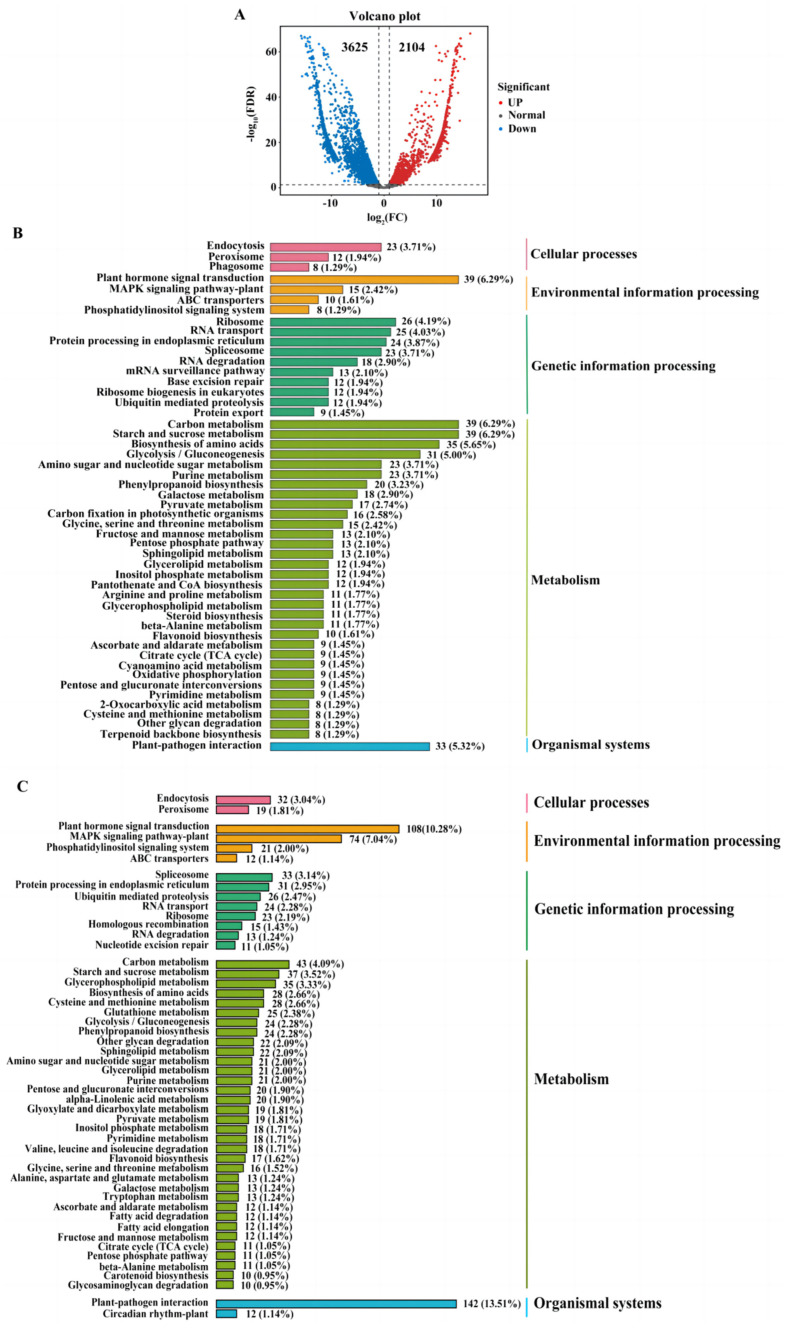
The DEG analysis in B73 and QR273. Notes: (**A**) Volcano plot of the DEGs between B73 and QR273; (**B**) The Kyoto Encyclopedia of Genes and Genomes (KEGG) classes for the upregulated genes; (**C**) The KEGG classes for the downregulated genes.

**Figure 6 ijms-24-10005-f006:**
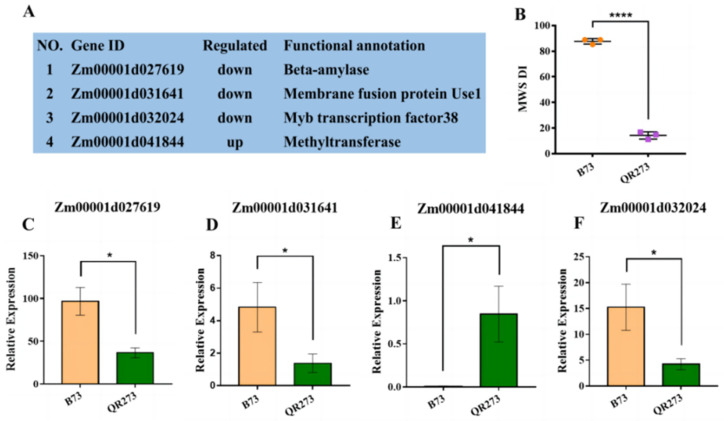
Candidate genes were identified by GWAS, resistance and transcriptome data. Notes: (**A**) Information on four candidate genes identified by GWAS; (**B**) MWS DI of QR273 and B73. Asterisks in (**B**) indicate statistically significant differences (****, *p * <  0.0001, Student’s *t*-test). (**C**–**F**) The relative expression levels of Zm00001d027619 (**C**), Zm00001d031641 (**D**), Zm00001d041844 (**E**) and Zm00001d032024 (**F**). Asterisks in (**C**–**F**) indicate statistically significant differences (*, *p* < 0.05, Student’s *t*-test).

**Figure 7 ijms-24-10005-f007:**
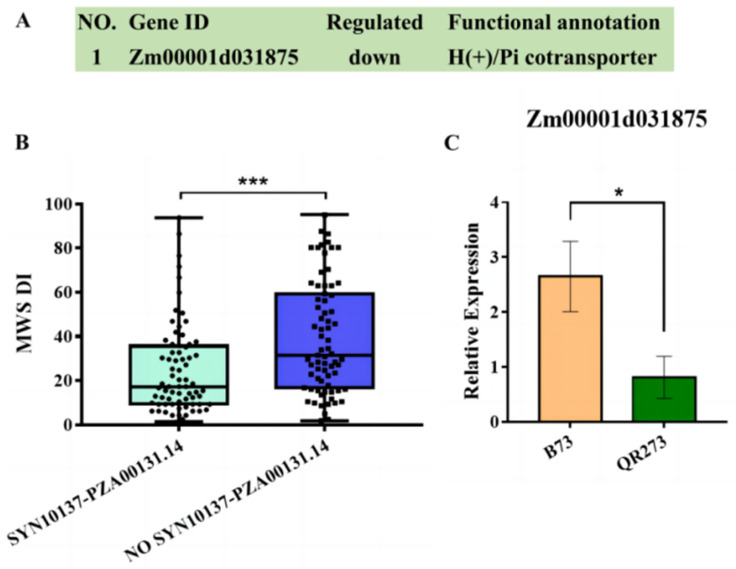
Candidate genes identification of the SYN10137-PZA00131.14 region. Notes: (**A**) Information on Zm00001d031875; (**B**) The MWS DI in the lines with and without the SYN10137-PZA00131.14 region; (**C**) Relative expression of Zm00001d031875. Asterisks in (**B**,**C**) indicate statistically significant differences (*, *p*  < 0.05, ***, *p*  <  0.001, Student’s *t*-test).

**Table 1 ijms-24-10005-t001:** The analysis of the variance for the MWS severity in a panel of 143 inbred lines across 3 environments.

Effect	Mean Square	*F*-Value
Genotype	3444.291	78.533 ***
Genotype × environment	777.392	17.725 ***
Error	43.858	

Notes: Asterisks in *F*-value indicate statistically significant differences (***, *p*  <  0.001, Friedman test).

**Table 2 ijms-24-10005-t002:** The common QTNs significantly associated with MWS under different environments.

QTN	Chr.	Position	Environment	Candidate Gene
SYNGENTA14387	1	9031311	Mojiang and joint environments	Zm00001d027619
PHM13619.5	1	22281738	Shilin and joint environments	Zm00001d028088
PZE-101040783	1	27552892	Shilin and Wenshan	
SYN11249	1	43111965	Shilin and joint environments	Zm00001d000328
PZE-101151153	1	194604587	Wenshan and joint environments	Zm00001d031641
SYN10891	1	195359337	Shilin and joint environments	Zm00001d031660
SYN10137	1	200556614	Shilin, Wenshan and joint environments	
PZA00131.14	1	203385274	Shilin, Wenshan and joint environments	
SYN37674	1	206370340	Shilin and joint environments	Zm00001d032024
PZE-102031753	2	14800008	Shilin and Wenshan	Zm00001d002549
PZE-103084298	3	135901771	Mojiang and joint environments	Zm00001d041844
ZM012464-0529	3	179012477	Shilin and joint environments	
SYN30108	4	231898350	Shilin and Wenshan	Zm00001d053432
PZE-108057528	8	101489430	Shilin, Wenshan and joint environments	Zm00001d010230
SYN4935	8	171846715	Shilin and joint environments	Zm00001d012665

**Table 3 ijms-24-10005-t003:** Classification of MWS disease resistance.

Scale	Area Occupied by Disease Spots
0	0
1	≤5%
3	6–10%
5	11–30%
7	31–50%
9	≥50%

**Table 4 ijms-24-10005-t004:** Evaluation standard of MWS resistance.

Scale	Disease Index (DI)	Resistance
0	DI = 0	Immunity (IM)
1	DI ≤ 10	High resistance (HR)
3	10 < DI ≤ 30	Resistance (R)
5	30 < DI ≤ 50	Moderate resistance (MR)
7	50 < DI ≤ 70	Susceptibility (S)
9	DI > 70	High susceptibility (HS)

## Data Availability

The data presented in this study are available on request from the corresponding author.

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
