# Peer review of "Integrated IBD Analysis, GWAS Analysis and Transcriptome Analysis to Identify the Candidate Genes for White Spot Disease in Maize"

_ijms, 2023, doi:10.3390/ijms241210005_

Round 1

Reviewer 1 Report

The article is written in an understandable language, a lot of data is covered. The authors competently built the study, substantiated the relevance of the chosen topic, presented the results of experiments and data analysis.

Nevertheless, there are some remarks regarding the design and content of the article.

Pay attention to the list of authors.

In figure 5, the inscriptions in the graphics are very illegible, the font is small, it is difficult to read.

There are quite a lot of outdated sources in the list of references, perhaps they should be replaced with newer ones.

Remarks are not grounds for rejecting the article, I recommend it for publication.

Author Response

Dear Reviewer,

On behalf of all the contributing authors, I would like to express our sincere appreciations of your letter and reviewers’ constructive comments concerning our article entitled “Integrated IBD analysis, GWAS analysis and transcriptome analysis to identify the candidate genes for White Spot Disease in maize” (Manuscript ID: ijms-2409912). These comments are all valuable and helpful for improving our article. According to the associate editor and reviewers’ comments, we have made accordingly modifications to our manuscript. Point-by-point responses to reviewer are listed below this letter.

1. Comment: Nevertheless, there are some remarks regarding the design and content of the article

1. Reply: We have made accordingly modifications to our manuscript. We hope our design will meet the requirements.

 2.Comment: Pay attention to the list of authors.

 2. Reply: We thank the reviewers for their reminders. We have corrected the order of the list of authors.

3. Comment: In figure 5, the inscriptions in the graphics are very illegible, the font is small, it is difficult to read.

3. Reply: We are sorry for not noticing the problem at the time. We have modified Figure 5 and other figures accordingly, and have re-uploaded.

4. Comment: There are quite a lot of outdated sources in the list of references, perhaps they should be replaced with newer ones.

4. Reply: We are sorry for this. We have revised some of the outdated references. However, there are some references that are more classic, so they are not replaced.

We sincerely hope that this revised manuscript has addressed all your comments and suggestions. We appreciated for reviewers’ warm work earnestly, and hope that the correction will meet with approval. Once again, thank you very much for your comments and suggestions.

Sincerely,
Dong Wang

Reviewer 2 Report

The authors in their manuscript entitled “Integrated IBD analysis, GWAS analysis and transcriptome analysis to identify the candidate genes for White Spot Disease in maize” utilize a combined approach based on genomic and transcriptomic analyses with the aim to locate plant genes with a role in WSP response/resistance.

The aim of research is clear, and the methodological approach and implementation are correct. The discussion adequately arguments on the results, and conclusions support the main findings. Though a revision regarding the experimental work per se is not needed, this (major revision) is required regarding presentation, and the discussion of certain results-parts. In specific:

I am missing a discussion/argumentation on the results presented in figure 1. The authors are encouraged to discuss the results providing an argumentation/bibliographic support on the finding that QR273 presents both HR and HS traits in descendants.

All figures should be easy to read in an A4 printed version of the manuscript. For example, figures 3, 4, 5 and 6 are very difficult to read once the manuscript is printed in paper. Figures 4 and 6 are nearly illegible, and figure 5 is actually illegible (please try it to print and read..). Please communicate with the editorial office to define if this is an original figure issue (authors) or an editorial issue (format) and resolve this. Please in all figures also increase the letter size since even in an enlarged figure (digital format) the letters are too small to understand. Figure legends are also poor in information. All figures should be able to stand alone regarding information given (being self-explanatory) and not needing to look back in the manuscript text to corelate. Thus, a full description of all abbreviations, axis, etc., and a relative explanatory text explaining the essence of the results in a concise way should be present in all figures. In the figure 2 legend also, please either remove the word “pairs” or add “and triple comparisons”. In figure 3 I could not count 30 hits (dots) with a p-value above 3. Please check this. Explain (either in the text or in the figure legend) what you expect/deduce from the Q-Q plot (figure 3b).

Please use KEGG information regarding the candidate genes identified and discuss (discussion section) if these are related (same biochemical pathways), and if so to what extend.

A revision of the manuscript for minor English language and expression mistakes is also needed to provide a more luminous text.

Minor corrections

Author Response

Dear Reviewer,

On behalf of all the contributing authors, I would like to express our sincere appreciations of your letter and reviewers’ constructive comments concerning our article entitled “Integrated IBD analysis, GWAS analysis and transcriptome analysis to identify the candidate genes for White Spot Disease in maize” (Manuscript ID: ijms-2409912). These comments are all valuable and helpful for improving our article. According to the associate editor and reviewers’ comments, we have made accordingly modifications to our manuscript. Point-by-point responses to reviewer are listed below this letter.

1. Comment: I am missing a discussion/argumentation on the results presented in figure 1. The authors are encouraged to discuss the results providing an argumentation/bibliographic support on the finding that QR273 presents both HR and HS traits in descendants.

1. Reply: We have added this content to the discussion section “3.1 Inheritance of MWS resistance”. QR273 is a tropical maize germplasm with good resistance, but its descendants exhibit varying degrees of resistance to MWS. Derivatives with resistance are mostly improved descendants between QR273 and tropical germplasm, while those with susceptibility are mostly the result of crossing with temperate germplasm. In the process of improving QR273 using temperate maize materials, due to different selection pressures, resistance probably was not emphasized in the selection of better resistant lines during breeding, or the resulting germplasm may have relatively poorer resistance due to a stronger genetic background from temperate maize material in the crossbreeding process.

2. Comment: All figures should be easy to read in an A4 printed version of the manuscript. For example, figures 3, 4, 5 and 6 are very difficult to read once the manuscript is printed in paper. Figures 4 and 6 are nearly illegible, and figure 5 is actually illegible (please try it to print and read..). Please communicate with the editorial office to define if this is an original figure issue (authors) or an editorial issue (format) and resolve this. Please in all figures also increase the letter size since even in an enlarged figure (digital format) the letters are too small to understand. Figure legends are also poor in information. All figures should be able to stand alone regarding information given (being self-explanatory) and not needing to look back in the manuscript text to corelate. Thus, a full description of all abbreviations, axis, etc., and a relative explanatory text explaining the essence of the results in a concise way should be present in all figures. In the figure 2 legend also, please either remove the word “pairs” or add “and triple comparisons”. In figure 3 I could not count 30 hits (dots) with a p-value above 3. Please check this. Explain (either in the text or in the figure legend) what you expect/deduce from the Q-Q plot (figure 3b).

2. Reply: Thank you for your comments. We have reworked the information related to all figures, removed the word “pairs” in the figure 2, and explained the Q-Q plot (figure 3b). We are very sorry for your unable to count 30 hits (dots) with a p-value above 3in figure 3, because of some QTNS are more similar in location and P value, so it's hard to find them in figure 3. You can clearly find 30 hits (dots) in Fig. S2.

3. Comment: Please use KEGG information regarding the candidate genes identified and discuss (discussion section) if these are related (same biochemical pathways), and if so to what extend.

3. Reply: We have added this content to the discussion section “3.3 Specific pathways response to diseases in maize”. Unfortunately, the relationship of these candidate genes was not found

4. Comment: A revision of the manuscript for minor English language and expression mistakes is also needed to provide a more luminous text.

4. Reply: Thank you for your suggestion, we have made modifications to the English language and expression mistakes of the manuscript.

We sincerely hope that this revised manuscript has addressed all your comments and suggestions. We appreciated for reviewers’ warm work earnestly, and hope that the correction will meet with approval. Once again, thank you very much for your comments and suggestions.

Sincerely,
Dong Wang

Reviewer 3 Report

The manuscript is based on a thorough study of white spot disease resistance in maize. Authors have a lot of diverse data to present and connect as a story in this manuscript. However, there is scope of further improvement. My suggestions include

1. Authors did not present ANOVA table analysis. Was there a significant genotype location interaction effect for DI? I suggest including it for clarity to readers. If genotype x location interaction is not significant, then only mean data of DI for MWS can be shown in the manuscript (as shown in the manuscript). But what if interaction comes out to be significant? Kindly check it and modify manuscript as per results.

2. Authors did not mention the validation of candidate genes using qRT-PCR in the methodology? Were the genes not validated? If not, then how come relative expression is shown in Fig 6 and 7? Is it based only on transcriptome data?

3. Authors did not mention about correction of p-values using Bonferroni correction method to avoid false positives in GWAS analysis/manhattan plots. The addition of this would bring clarity to readers.

4. Suggested paper may be useful to support the candidate genes/QTNs-

Meta-QTL analysis for mining of candidate genes and constitutive gene network development for fungal disease resistance in maize (Zea mays L.)

Authors need to improve the language of the manuscript in terms of grammar/spelling mistakes to bring in more clarity in sentences (please see highlighted sections). Authors need to carefully use abbreviations, only if they are used repeatedly in manuscript (See highlighted texts in manuscript).

Author Response

Dear Reviewer,

On behalf of all the contributing authors, I would like to express our sincere appreciations of your letter and reviewers’ constructive comments concerning our article entitled “Integrated IBD analysis, GWAS analysis and transcriptome analysis to identify the candidate genes for White Spot Disease in maize” (Manuscript ID: ijms-2409912). These comments are all valuable and helpful for improving our article. According to the associate editor and reviewers’ comments, we have made accordingly modifications to our manuscript. Point-by-point responses to reviewer are listed below this letter.

1. Comment: Authors did not present ANOVA table analysis. Was there a significant genotype location interaction effect for DI? I suggest including it for clarity to readers. If genotype x location interaction is not significant, then only mean data of DI for MWS can be shown in the manuscript (as shown in the manuscript). But what if interaction comes out to be significant? Kindly check it and modify manuscript as per results.

1. Reply: We have presented the ANOVA table, see Table 1. And the significant differences among genotypes for genotypes × environment were observed for MWS severity.

2. Comment: Authors did not mention the validation of candidate genes using qRT-PCR in the methodology? Were the genes not validated? If not, then how come relative expression is shown in Fig 6 and 7? Is it based only on transcriptome data?

2. Reply: In this study, qRT-PCR was done, and I apologize for forgetting to mention it in the manuscript, it has been added now. The expression trends of the five genes were consistent with the transcriptome analysis results.

3. Comment: Authors did not mention about correction of p-values using Bonferroni correction method to avoid false positives in GWAS analysis/manhattan plots. The addition of this would bring clarity to readers.

3. Reply: We are sorry for this. The GWAS was operated in the R package GAPIT using the MLM, with population structure and pair kinship as covariates index to correct for false positive associations between individual loci and phenotypes.

 4. Comment: Suggested paper may be useful to support the candidategenes/QTNs-Meta-QTL analysis for mining of candidate genes and constitutive gene network development for fungal disease resistance in maize (Zea maysL.)

4. Reply: Thanks to the reviewers for their comments, we will try to contribute in this work.

We sincerely hope that this revised manuscript has addressed all your comments and suggestions. We appreciated for reviewers’ warm work earnestly,and hope that the correction will meet with approval. Once again, thank you very much for your comments and suggestions.

Sincerely,
Dong Wang

Round 2

Reviewer 2 Report

No further comments

Author Response

Dear Reviewer,

On behalf of all the contributing authors, I would like to express our sincere appreciations of your letter and reviewers’ constructive comments concerning our article entitled “Integrated IBD analysis, GWAS analysis and transcriptome analysis to identify the candidate genes for White Spot Disease in maize” (Manuscript ID: ijms-2409912). These comments are all valuable and helpful for improving our article. Once again, thank you very much for your comments and suggestions.

Sincerely,
Dong Wang

Reviewer 3 Report

Suggestions has been incorporated but one major issue remains. As authors respond "1. ReplyWe have presented the ANOVA table, see Table 1. And the significant differences among genotypes for genotypes × environment were observed for MWS severity."

Then the GWAS for MWS resistance should be done separately for each environment. Currently, it is based on the mean of three environments, but it is inappropriate as genotypes performance/ranking for MWS resistance changes across environments as indicated from significant G x E. It is equivalent to creating biases as one genotype showing high resistance in one environment may be extremely susceptible/moderately susceptible in another environment and averaging it puts in an intermediate resistance.

Authors should perform separate GWAS for each environment. It will be more appropriate and will give the opportunity to explore stable associations/SNPs for MWS resistance across environments. 

Minor editing required

Author Response

Dear Reviewer,

On behalf of all the contributing authors, I would like to express our sincere appreciations of your letter and reviewers’ constructive comments concerning our article entitled “Integrated IBD analysis, GWAS analysis and transcriptome analysis to identify the candidate genes for White Spot Disease in maize” (Manuscript ID: ijms-2409912). These comments are all valuable and helpful for improving our article. According to the associate editor and reviewers’ comments, we have made accordingly modifications to our manuscript. Point-by-point responses to reviewer are listed below this letter.

1. Comment: Suggestions has been incorporated but one major issue remains. As authors respond "1. Reply: We have presented the ANOVA table, see Table 1. And the significant differences among genotypes for genotypes × environment were observed for MWS severity." Then the GWAS for MWS resistance should be done separately for each environment. Currently, it is based on the mean of three environments, but it is inappropriate as genotypes performance/ranking for MWS resistance changes across environments as indicated from significant G x E. It is equivalent to creating biases as one genotype showing high resistance in one environment may be extremely susceptible/moderately susceptible in another environment and averaging it puts in an intermediate resistance. Authors should perform separate GWAS for each environment. It will be more appropriate and will give the opportunity to explore stable associations/SNPs for MWS resistance across environments.

1. Reply: Thank you for your suggestion, we have performed GWAS for each environment separately. The GWAS results showed that, 73, 25, and 31 quantitative trait nucleotides (QTNs) (P< 0.001) were significantly associated with MWS under environment of Shilin (Table S3, Fig. 3A, Fig. S2), Mojiang (Table S4, Fig. 3B, Fig. S3), and Wenshan (Table S5, Fig. 3C, Fig. S4), respectively. Overall, a total of 15 common QTNs were found after the comparison of results under different environments (Table 2). This included QTNs previously obtained by GWAS using the average value of MWS DI among three environments. For example, SYN10137 and PZA00131.14 were co-identified under Shilin, Wenshan. Also, a relevant analysis was conducted based on these results.

We sincerely hope that this revised manuscript has addressed all your comments and suggestions. We appreciated for reviewers’ warm work earnestly, and hope that the correction will meet with approval. Once again, thank you very much for your comments and suggestions.

Sincerely,

Dong Wang

Round 3

Reviewer 3 Report

Comments have been well addressed by the authors.